# Factors associated with early sexual debut among adolescents and youth in Mozambique: A geo-additive survival analysis of the Mozambique 2021 AIDS indicator survey

Rachid Muleia[1]*, Shelsea Luís Damião[2], Áuria Ribeiro Banze[1], Cynthia Semá Baltazar[1], Isaac Akpor Adjei[3]

1 Instituto Nacional de Saúde, Maputo, Mozambique, 2 Universidade Eduardo Mondlane, Maputo, Mozambique, 3 Kwame Nkrumah University of Science and Technology, Kumasi, Ghana

* rachid.muleia@ins.gov.mz

**Data availability statement:** The data used is freely available at https://github.com/RachidMuleia/sexual_debut.git.

## Abstract

Engaging in sexual activity before the age of 15, known as early sexual debut, is a significant public health concern due to its association with increased risks of sexually transmitted infections, unintended pregnancies, and reduced educational achievements. This research investigates the factors influencing early sexual initiation among adolescents and youth aged 15 to 24 years in Mozambique, utilizing data of 5,283 individuals from the 2021 Mozambique AIDS indicator survey. Geoadditive models for censored time-to-event outcomes were employed to explore regional disparities and key determinants of the time to sexual debut, aiming to inform region-specific public health interventions and policies. The model was fitted using Poisson approximation within the framework of mixed model formulation for generalized additive models. The results revealed that approximately 80% of individuals have engaged in sexual activity, with 18.9% reporting having had sex before the age of 15. Among women, the percentage was 16.3%, while among men, it was 26.7%. Regional disparities were also observed, with the northern region exhibiting a higher prevalence of sexual debut before the age of 15 (26.7%). The median age at which individuals initiate sexual activity was estimated to be 16 years. After adjusting for significant covariates, the findings indicated a higher incidence of early sexual debut in the northern and southern regions of Mozambique, with notable gender differences. Age, occupation, alcohol consumption, illicit drug use, and living with parents were statistically significant factors associated with the age of sexual debut. These results highlight the need of addressing both individual behaviors and broader structural determinants—such as education and socio-economic conditions—to delay sexual initiation and reduce adverse health outcomes among Mozambican adolescents and youth. This study adds to the limited geospatial literature on sexual behavior in Mozambique.

**Funding:** The author(s) received no specific funding for this work.

**Competing interests:** The authors have declared that no competing interests exist.

## Introduction

Early sexual debut, defined as initiating sexual activity before the age of 15, is a significant public health concern associated with various adverse outcomes, including increased risk of sexually transmitted infections (STIs), unintended pregnancies, and lower educational attainment. Studies point out that early sexual debut can also lead to engagement in risky sexual behavior both in adolescence and adulthood, such as multiple sexual partners and incorrect or inconsistent condom use, which may eventually lead to HIV/AIDS. Among females, early sexual debut can also lead to childbirth injuries such as fistula or even death, whereas, among males, studies show that it can also lead to arousal problems, erectile dysfunction, and orgasm problems [1,2]. A recent study conducted with 12 demographic health surveys (DHS) from East African countries shows that rates of early sexual debut vary across countries among males, ranging from 3.5% to 35.7%, with Mozambique having the highest rate [3]. Another study, which analyzed DHS data from 33 sub-Saharan African countries, found a staggering pooled prevalence of early sexual debut among females of 71.0% [4].

The high rates and variation of early sexual debut across countries can be explained by a wide range of individual socio-demographic and structural level factors. At an individual level, several studies highlight education, sex, socio-economic status, alcohol drinking, media exposure, comprehensive knowledge of HIV and knowledge of family planning to be linked to the timing of sexual debut [5]. A recent study by [3] revealed that as the level of education increases, the risk of engaging in early sexual debut diminishes, a phenomenon that can probably be explained by the fact that more educated individuals are more aware of the negative effects of early sex debut [6]. Moreover, structurally, the timing at which individuals initiate their sexual activity may be determined by gender expectations, access to healthcare service conflicts and ethnicity. For instance, in some societies, masculinity often includes stereotypes like the "sex role model", which suggests men have uncontrollable sexual needs, viewing dominance over women as normal, and see multiple sexual partners as a sign of sexual prowess. Consequently, young men may engage in early sexual debut as a way to prove themselves [7,8].

In many sub-Saharan African countries, early sexual debut is a prevalent issue, contributing to the region's high rates of HIV/AIDS and other health challenges. In 2023 alone, approximately 360,000 young people aged 15 to 24 were newly infected with HIV, with 140,000 of these infections occurring among adolescents aged 15 to 19 [9]. Furthermore, each year in sub-Saharan Africa, around 14 million unwanted pregnancies occur, with 44% being among adolescent and young women and 13% childbearing before 18 [10,12]. Therefore, understanding the factors associated with time to sexual debut is crucial for developing effective interventions to delay sexual initiation and reduce associated risks.

Mozambique, like many countries in sub-Saharan Africa, faces a high burden of HIV/AIDS, with adolescents and young adults being particularly vulnerable. According to the latest Mozambique Population-based HIV Impact Assessment (INSIDA 2021), sexual behaviors, including the timing of sexual debut, play a significant role in the transmission dynamics of HIV among the youth population [13]. Despite the growing body of literature on sexual behavior and health outcomes in Mozambique, there is limited geospatial analysis that explores the regional variations and determinants of the time to sexual debut among adolescents and youth.

Studies suggest that as education levels rise, the likelihood of early sexual initiation tends to decrease, likely because education fosters greater awareness of the potential consequences of early sexual activity. Conversely, youth from socioeconomically disadvantaged backgrounds or those exposed to alcohol and substance use are more prone to early sexual

initiation [3,6]. However, there is limited research on how these factors vary across Mozambique's diverse regions. Given the country's regional disparities in social norms and health access, this study aims to fill this gap by conducting a geospatial analysis of factors associated with time to sexual debut among adolescents and youth in Mozambique, utilizing data from the Mozambique 2021 AIDS survey. By examining the socio-demographic, behavioural, and geographic factors that contribute to early sexual debut, this study seeks to provide insights.

## Materials and methods

### Study design and sample

The current study was based on the latest Mozambique Population-based HIV Impact Assessment, INSIDA 2021, a cross-sectional household-based survey designed to assess the burden of HIV disease and the impact of the health sector response to national HIV epidemics. The INSIDA 2021 is a cross-sectional complex survey with a stratified four-stage cluster sampling design targeting the adult population aged 15 years or older, and collects a wide range of information on knowledge, attitude and risk behaviour related to HIV infection, socio-demographic and cultural factors, etc. In the first stage, a sample of control areas (CA), a sampling unit comprising three to four enumeration areas (EA), was selected from the 2017 population and housing census frame. This was followed by an equal probability sample of one EA within each CA. After selecting the EA, dwelling units within each EA were selected in the third stage. The dwelling units were sampled with probability proportional to the number of eligible households they contained. Subsequently, one household within each of the selected dwelling units was randomly chosen to participate in the study, and finally, all eligible persons within the selected household were interviewed. In addition to socio-demographic, behavioural, and HIV data, the surveys also gathered geographic coordinates of the enumeration areas (EAs) from which the households were sampled. Further details on the sample design are fully described elsewhere [13]. The survey data set is publicly available at https://phia.icap.columbia.edu/surveys/ upon reasonable request. For the current study, we included 5,283 adolescents and youth aged 15 to 24 years who consented to participate.

### Study measures

**Outcome variable.** To investigate the geographical distribution of early sexual debut and its associated factors, we considered the age at first sexual experience as the outcome variable. This outcome variable was treated as a censored time-to-event variable. Consequently, survival time was defined as the duration from birth to the first sexual encounter, measured in years at the time of the survey. For individuals who have engaged in sexual activity, their age at first sexual experience was recorded. In contrast, for those who have not yet had sex, their current age at the time of the survey was noted as their survival time. Thus, the event of interest is the initiation of sexual activity, specifically referring to vaginal intercourse. Individuals who reported never having had sex by the time of the survey were censored, with their current age serving as the censoring time.

**Explanatory variables.** The explanatory variables considered in this study were selected based on existing literature [14–16] and their availability in the INSIDA survey data. These variables encompass socio-demographic factors, including the participant's age (15-19 years, 20-24 years), sex (female, male), residence (urban, rural), education level (no education, primary, secondary, and higher), region (south, center, north), marital status (single, married, divorced/widowed), occupation (unemployed, employed), and wealth index (low, middle, high). The wealth index is an aggregate measure of a household's cumulative living standard,

derived from data on household assets and calculated using Principal Component Analysis (PCA). The PCA scores are categorized into five wealth quintiles, which were subsequently reclassified into three groups: low (first two quintiles), middle (third quintile), and high (last two quintiles). Additionally, we considered behavioural factors, specifically alcohol consumption (yes, no) and illicit drug use in the last 12 months (yes, no). We also included whether the participant had ever stayed away from home for more than one month (yes, no), had ever discussed sex with parents (yes, no), had ever discussed HIV with parents, and whether the participant had ever attended any intervention or HIV prevention program.

## Statistical analysis and estimation procedure

**Exploratory data analysis.** We performed descriptive univariate and bivariate analyses to provide a comprehensive understanding of the study variables. The univariate analysis was conducted to illustrate the distribution of study participants across socio-demographic variables. In contrast, bivariate analysis was employed to present the prevalence of early sexual debut by explanatory variables. The Pearson Chi-square test of independence was utilized to examine the relationship between early sexual debut and socio-demographic variables. Additionally, Kaplan-Meier survival curves were generated to estimate the survival probability of experiencing sexual debut at different ages [17], followed by a log-rank test to compare differences in survival curves across variable categories.

**Geoadditive model for survival outcome.** This study relies on flexible additive models for survival outcomes reduced to a generalized linear mixed model to model the spatial distribution of the hazard of sexual debut among adolescents and youth. Our methodology is based on the work of [18], who built upon the research of [19] to approximate the Cox regression model for failure time outcomes using a Poisson regression with an offset. The data comprises a random sample, $(T_i, C_i)$, where $T_i$ is a non-negative continuous time representing the duration until a specific outcome is measured from a suitable starting point (in our study, this is the time from birth to the first sexual experience) and $C_i$ represents the censored time. The observed duration can be represented by $t_i = \min(T_i, C_i)$. Thus, the observed data can be represented by

$$(t_i, \delta_i, \mathbf{x}_i, \mathbf{s}_i), \quad i = 1, \dots, n,$$

where $\delta_i$ denotes an indicator for censoring, $\mathbf{x}_i$ represents a row vector of covariates and $\mathbf{s}_i$ stands for the geographical location defined by the latitude and longitude values for the $i^{\text{th}}$ individual. With that said, we can assume that the data are generated from a distribution with a hazard function specified as follows:

$$\lambda_i(t|\mathbf{x}_i, \mathbf{s}_i) = \lambda_0 \exp\left(\boldsymbol{\beta}^T \mathbf{x}_i + S(\mathbf{s_i})\right), \tag{1}$$

where $\lambda_i(t|\mathbf{x}_i, \mathbf{s}_i)$, $t>0$, is the hazard function at time $t$ for the $i^{\text{th}}$ individual, while $\lambda_0$ denotes the unspecified baseline hazard, and $\boldsymbol{\beta}$ a vector of parameters related to the different covariates. In equation (1), the component $S(\mathbf{s}_i)$ represents an unspecified bivariate smooth function. This is a geographical component and acts as a surrogate for all unmeasured spatially referenced covariates. Note that the log hazard function is an additive function of covariates. Furthermore, the hazard ratio between two subjects is assumed to be constant over time.

Similar to [18], we also rely on the approach of [19], which approximates the hazard function (1) to a Poisson regression model through martingale theory:

$$E(N_i|\mathbf{x}_i, \mathbf{s}_i) = \exp\left(\boldsymbol{\beta}^T \mathbf{x}_i + S(\boldsymbol{s_i})\right) \int_0^\infty Y_i(v)\lambda_0(v)dv, \tag{2}$$

where $N_i(t) = I\big[T_i \le t, \delta = 1\big]$ and $Y_i(t) = I\big[T_i > t\big]$ for $i = 1, \dots, n$ in (2) represent the counting process evaluated at $t = \infty$. It should be noted that the terms $\int_0^\infty Y_i(v)\lambda_0(v)dv$ are unknown. Nevertheless, [19] assume these to be equivalent to the cumulative baseline hazards in the standard Cox model, $\hat{\Lambda}_i$, which translates (2) into a Poisson regression model with known offsets:

$$E\big(N_i|\mathbf{x}_i, \mathbf{s}_i\big) = \exp\left(\boldsymbol{\beta}^T\mathbf{x}_i + S(\mathbf{s_i})\right)\hat{\Lambda}_i \tag{3}$$

In equation (3), the geographical component $S(\boldsymbol{s}_i)$ is taken to be a penalized spline function of the form:

$$S(\mathbf{s}_i) = \mathbf{s}_i\boldsymbol{\beta}_s + \sum_{k=1}^{K_s} u_k^s z_k^s(\mathbf{s}_i),$$

where $u_1^s, \dots, u_k^s$ are treated as random coefficients, distributed independently and identically, i.e, $u_k^s \sim N(0, \sigma_s^2)$ and $\{z_1^s, \dots, z_{K_s}^s\}$ is a set of $K_s$ bivariate spline basis functions. Treating $u_k$ as random coefficients enables one to use the connection between a smoothing spline and a mixed model [20,21]. Consequently, the equation (3) can be expressed as follows:

$$N_i|\mathbf{u} \sim \text{Poisson}\big[\exp\{\big(\mathbf{X}\boldsymbol{\beta} + \mathbf{Zu}\big)_i\hat{\Lambda}_i\big], \quad \mathbf{u} \sim \text{N}(\mathbf{0}, \sigma_u^2\boldsymbol{I}), \tag{4}$$

where $\mathbf{X} = \begin{bmatrix}\mathbf{x}_i & \mathbf{s}_i\end{bmatrix}_{1\le i\le n}$ and $\mathbf{Z}$ is the design matrix containing the bivariate spline basis functions, which is defined via radial spline bases functions. Note that as the baseline hazard function already accounts for the intercept, this is not included in the design matrix $\mathbf{X}$ in (4).

Parameter estimation of model (4) can be tailored within the mixed model framework using penalized quasi-likelihood as described in [22]. Nevertheless, for this to be possible $\mathbf{Z}$ should be defined as follows:

$$\mathbf{Z} = \big[z_s^k(\mathbf{s}_i)\big]_{1\le i\le n, 1\le k\le K_s} = \big[g_\tau(\mathbf{s}_i - \boldsymbol{\kappa}_k)\big]_{1\le i\le n, 1\le k\le K_s} \cdot \big[g_\tau(\boldsymbol{\kappa}_k - \boldsymbol{\kappa}_k')\big]_{1\le k\le K_s, 1\le k'\le K_s}^{-1/2},$$

where $g_\tau$ is a valid generalized covariance function used in kriging [23,24]. Herein, we consider covariance functions that are members of the Matérn family of covariance functions, namely, the exponential $(\exp(-\|\mathbf{h}\|/\tau))$, gaussian $(\exp(-\|\mathbf{h}\|^2/\tau^2))$, Matérn $(\exp(-\|\mathbf{h}\|/\tau)(1 + \|\mathbf{h}\|/\tau))$, spherical $((1 - 1.5\|\mathbf{h}\|/\tau + 0.5\|\mathbf{h}\|^3/\tau^3)I_{\|\mathbf{h}\|<\tau})$, circular $(1 - \frac{2}{\pi}(\vartheta\sqrt{1 - \vartheta^2} + \arcsin\vartheta), \text{ with } \vartheta = \min(\frac{\|\mathbf{h}\|}{\|}\tau, 1))$, and inverse quadratic $(1/\sqrt{1 + \|\mathbf{h}\|^2/\tau})$ covariance functions. Observe that for model (4) to be couched within the mixed model framework, the range parameter $\tau$ in the covariance functions needs to be set to a specific value. [25] set the range parameter to be equal to the maximum distance between the observation, that is, $\tau = \max_{1\le i,j\le n} \|\mathbf{s}_i - \mathbf{s}_j\|$. Herein, $\tau$ is estimated from the data, following a two-stage iterative likelihood-based estimation process proposed by [26] for Gaussian response and further improved for non-Gaussian outcome by [27]. It is worth stressing that the complex survey design is accounted for by using a weighted likelihood, with weights corresponding to the inverse of the sampling probabilities. To determine the most suitable covariance function, we employed Akaike's Information Criterion (AIC), the corrected AIC, and the Bayesian Information Criterion (BIC) [28,29]. Models exhibiting lower AIC and BIC values are deemed to offer a superior fit. The corrected AIC was utilized because the traditional AIC inadequately accounts for the degrees of freedom in the unspecified bivariate function and overlooks the

spatial dependence present in the data [30]. This exercise led to the spherical covariance function for the analysis. Furthermore, with regard to the number of knots, we followed the Ruppert's guidelines, which suggest taking the number of knots as $K = \max\{20, \min(n/4, 150)\}$, while the selection of the knots' location was done using a space-filling algorithm [31–33].

All analyses in this study were performed using the R statistical software version 4.4.1 [34]. Statistical significance was assessed at a significance level of 5%. The R code to fit the geoadditive model for survival outcome is available in a Github repository for public access (https://github.com/RachidMuleia/sexual_debut.git).

## Ethical considerations

This study used data from the Mozambique National AIDS Indicator Survey (INSIDA 2021), conducted as part of the Population-based HIV Impact Assessment (PHIA) project. This survey was implemented by the Instituto Nacional de Saúde (INS) in collaboration with the Ministry of Health (MISAU) and with technical assistance from ICAP at Columbia University, with the objective of assessing HIV prevalence and other health indicators in Mozambique.

All protocols, consent forms, screening forms, referral forms, recruitment materials, and questionnaires for INSIDA 2021 were reviewed and approved by national regulatory and ethical bodies, including the Mozambique National Bioethics Committee for Health (CNBS), local institutional review boards (IRBs) where available, and the IRBs of Columbia University Medical Center, Westat, and the Centers for Disease Control and Prevention (CDC) [13].

Participants were informed of the study objectives, procedures, potential risks, and benefits, and provided informed consent prior to participation. Data were anonymized to ensure participant privacy and confidentiality, adhering to best practices for data security and privacy in population health research.

The authors of this paper received authorization from the PHIA to use de-identified data exclusively for secondary analysis. This analysis adheres to the ethical principles of confidentiality, autonomy, and non-maleficence, in alignment with the Declaration of Helsinki and the ethical guidelines of the PHIA project

## Results

### Socio-demographic profile of the study participants

Table 1 summarizes the participants who engaged in sexual activity. Of the 5,283 youth aged 15–24, approximately 80% reported to have engaged in sexual activity. Notably, 18.9% of the participants reported having had sexual intercourse before the age of 15. The results further show that the rate of early sexual debut is significantly higher among male individuals (21.6 % vs 16.3%, $p = 0.001$).

We also note that early sexual debut is highly prevalent in the northern region, with a rate of 26.7%, followed by the central and southern regions with prevalence of 16.0% and 14.1%, respectively. Similarly, when disaggregated by sex, the pattern remains consistent. Furthermore, there is a significantly higher prevalence of early sexual debut among individuals in rural areas (20.9% vs. 16.5%, $p = 0.014$). Among females, similarly, the prevalence of early sexual debut is significantly higher in rural areas (20.6% vs 10.9%, $p<0.001$), whereas, among males, the prevalence is statistically equivalent in urban and rural areas (22.3% vs 21.1%, $p = 0.597$).

The results further indicate that, in general, early sexual debut is significantly more common among individuals with a lower socio-economic status compared to those with a higher

**Table 1. Distribution of early sexual debut (sex before 15 years old) prevalence by demographic factors.**

| Variables | N (%) | Overall Yes, n(%) | No, n(%) | P-value | Female Yes, n(%) | No, n(%) | P-value | Male Yes, n(%) | No, n(%) | P-value |
|---|---|---|---|---|---|---|---|---|---|---|
| **Age** | | | | | | | | | | |
| 15-19 | 2579 (54.2) | 410 (18.7) | 2069 (81.3) | | 188 (16.7) | 1164 (83.3) | | 222 (20.8) | 905 (79.2) | |
| 20-24 | 2704 (45.8) | 438 (19.1) | 2066 (80.9) | 0.802 | 205 (15.8) | 1241 (84.2) | 0.627 | 233 (22.8) | 825 (77.2) | 0.333 |
| **Sex** | | | | | | | | | | |
| Male | 2320 (48.5) | 455 (21.6) | 1730 (78.4) | 0.001 | – | – | – | – | – | – |
| Female | 2963 (51.5) | 393 (16.3) | 2405 (83.7) | 0.001 | – | – | – | – | – | – |
| **Region** | | | | | | | | | | |
| North | 1443 (32.4) | 348 (26.7) | 972 (73.3) | | 176 (25.7) | 558 (74.3) | | 172 (27.8) | 414 (72.2) | 0.001 |
| Center | 1974 (39.9) | 251 (16) | 1604 (84) | | 120 (13.9) | 907 (86.1) | | 131 (18.2) | 697 (81.8) | 0.001 |
| South | 1866 (27.7) | 249 (14.1) | 1559 (85.9) | <0.001 | 97 (9.4) | 940 (90.6) | <0.001 | 152 (19.4) | 619 (80.6) | 0.001 |
| **Residence** | | | | | | | | | | |
| Urban | 2508 (44.4) | 375 (16.5) | 2038 (83.5) | | 142 (10.9) | 1211 (89.1) | | 233 (22.3) | 827 (77.7) | |
| Rural | 2775 (55.6) | 473 (20.9) | 2097 (79.1) | 0.014 | 251 (20.6) | 1194 (79.4) | <0.001 | 222 (21.1) | 903 (78.9) | 0.597 |
| **Wealth index** | | | | | | | | | | |
| Low | 1485 (31) | 291 (23.3) | 1061 (76.7) | | 160 (23.7) | 593 (76.3) | | 131 (23) | 468 (77) | |
| Middle | 837 (16.5) | 134 (18.9) | 651 (81.1) | | 72 (18.0) | 380 (82.0) | | 62 (20.) | 271 (80.0) | |
| High | 2941 (52.5) | 419 (16.4) | 2409 (83.6) | 0.003 | 160 (11.6) | 1425 (88.4) | <0.001 | 259 (21.4) | 984 (78.6) | 0.696 |
| **Education level** | | | | | | | | | | |
| No Education | 479 (10.4) | 104 (27.4) | 306 (72.6) | | 71 (29.6) | 202 (70.4) | | 33 (24) | 104 (76) | |
| Primary | 2101 (41.9) | 392 (21.8) | 1571 (78.2) | | 209 (21.2) | 913 (78.8) | | 183 (22.5) | 658 (77.5) | |
| Secondary/Higher | 2697 (47.7) | 352 (14.7) | 2253 (85.3) | <0.001 | 113 (8.5) | 1285 (91.5) | <0.001 | 239 (20.7) | 968 (79.3) | 0.562 |
| **Occupation** | | | | | | | | | | |
| Unemployed | 3706 (71.7) | 563 (18.2) | 2938 (81.8) | | 327 (16.9) | 1922 (83.1) | | 236 (20.2) | 1016 (79.8) | |
| Employed | 1569 (28.3) | 285 (20.6) | 1191 (79.4) | 0.141 | 66 (13.6) | 481 (86.4) | 0.097 | 219 (23.9) | 710 (76.1) | 0.048 |
| **Marital status** | | | | | | | | | | |
| Single | 3114 (60.8) | 436 (16.3) | 2554 (83.7) | | 116 (10.2) | 1225 (89.8) | | 320 (20.5) | 1329 (79.5) | |
| Married | 1791 (32.2) | 338 (22.8) | 1301 (77.2) | | 225 (21.5) | 976 (78.5) | | 113 (25.9) | 325 (74.1) | 0.067 |
| Divorced/Widowed | 378 (7) | 74 (24) | 277 (76) | <0.001 | 52 (24.1) | 202 (75.9) | <0.001 | 22 (23.6) | 75 (76.4) | 0.067 |
| **Alcohol consumption** | | | | | | | | | | |
| Yes | 1041 (17.8) | 218 (24.7) | 767 (75.3) | | 56 (18.6) | 340 (81.4) | | 162 (28.0) | 427 (72.0) | |
| No | 4226 (82.2) | 627 (17.6) | 3355 (82.4) | <0.001 | 334 (15.9) | 2056 (84.1) | 0.210 | 293 (19.7) | 1299 (80.3) | <0.001 |
| **Ever lived away from home for more than one month** | | | | | | | | | | |
| Yes | 1229 (22.1) | 208 (19.0) | 966 (81.0) | | 68 (13.6) | 499 (86.4) | | 140 (23.4) | 467 (76.6) | |
| No | 4036 (77.9) | 637 (18.8) | 3158 (81.2) | 0.892 | 323 (16.9) | 1901 (83.1) | 0.135 | 314 (21.0) | 1257 (79.0) | 0.318 |
| **Used illicit drugs in the last 12 months** | | | | | | | | | | |
| Yes | 94 (1.9) | 21 (22.1) | 66 (77.9) | | 4 (49.8) | 8 (50.2) | | 17 (19.0) | 58 (81.0) | |
| No | 5180 (98.1) | 826 (18.8) | 4061 (81.2) | 0.539 | 388 (16.1) | 2393 (83.9) | 0.002 | 438 (21.8) | 1668 (78.2) | 0.613 |
| **Ever talked with parents about sex** | | | | | | | | | | |
| Yes | 917 (16.1) | 141 (18.5) | 739 (81.5) | | 64 (14.9) | 467 (85.1) | | 77 (23.2) | 272 (76.8) | |
| No | 4345 (83.9) | 702 (18.9) | 3381 (81.1) | 0.822 | 324 (16.4) | 1929 (83.6) | 0.527 | 378 (21.5) | 1452 (78.5) | 0.526 |
| **Ever talked with parents about HIV/AIDS** | | | | | | | | | | |
| Yes | 1039 (18.4) | 140 (20.0) | 711 (80.0) | | 59 (14.3) | 474 (85.7) | | 81 (28.0) | 237 (72.0) | |
| No | 4228 (81.6) | 705 (19.3) | 3274 (80.7) | 0.134 | 332 (17.3) | 1849 (82.7) | 0.021 | 373 (21.4) | 1425 (78.6) | 0.497 |
| **Ever attended any prevention program** | | | | | | | | | | |
| Yes | 4910 (96.3) | 801 (19.2) | 3827 (80.8) | | 368 (16.4) | 2226 (83.6) | | 433 (22.1) | 1601 (77.9) | |
| No | 217 (3.7) | 28 (15.3) | 185 (84.7) | 0.309 | 13 (13.4) | 113 (86.6) | 0.543 | 15 (17.5) | 72 (82.5) | 0.399 |
| **Overall** | | 848 (18.9) | 4135 (81.1) | – | – | – | – | – | | |

socio-economic status (23.3% vs. 16.4%, *p* = 0.003). A similar pattern is observed among women (23.7% vs 11.6%, *p*<0.001). From Table 1, it is also apparent that early sexual debut is significantly higher among individuals with no education compared to those with a secondary education level (27.4% vs 14.7%, *p*<0.001). This pattern is also evident among females

(29.6% vs 8.5%, *p*<0.001). The results also show that, overall, early sexual debut is significantly more frequent among individuals who reported alcohol consumption (24.7% vs. 17.6%, *p*<0.001).

## Bivariate analysis using Kaplan - Meier survival analysis

The relationship between demographic factors and the timing of sexual activity initiation is analyzed using Kaplan-Meier survival curves stratified by sex. The overall Kaplan-Meier survival curve for all participants indicates that by age 15, over a quarter had experienced sexual intercourse at least once (Fig 1). Additionally, the median age at which individuals first engage in sexual activity is estimated to be 16 years. Furthermore, Fig 1 demonstrates that the survival curves for the first sexual experience are not significantly different between women and men.

**Kaplan-Meier survival analysis for female individuals.** The sex-stratified analysis in Fig 2 reveals that, among women, the survival curves for the timing of sexual activity initiation do not significantly differ between those aged 15–19 and 20–24. The Kaplan-Meier curve shows that women from rural areas are more likely to begin sexual activity earlier than those from urban areas as they age. Additionally, the results indicate that women from the northern region have a higher likelihood of earlier sexual debut as they grow older compared to those from the central and southern regions. Fig 2 also shows that before age 18, the probability of early sexual debut is higher among women with lower socio-economic status compared to those with higher socio-economic status. Moreover, the results suggest that women who consume alcohol have a higher probability of initiating sexual activity earlier as they age. Similarly, women who reported using illicit drugs in the past 12 months also have a higher probability of early sexual debut.

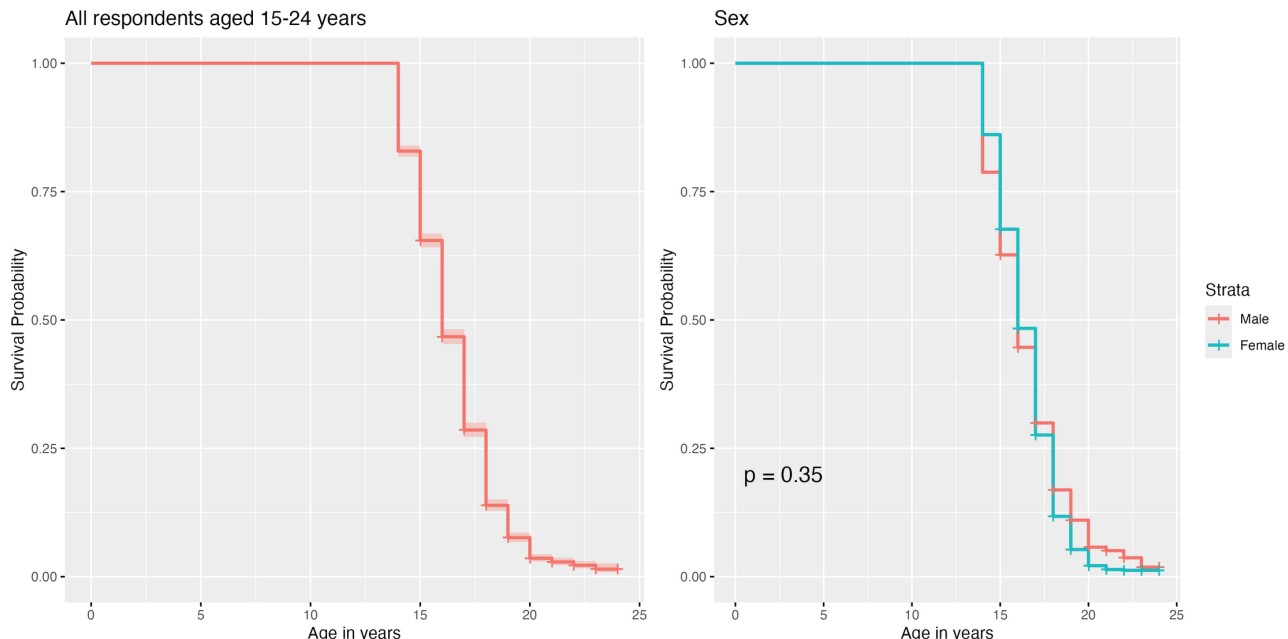

**Fig 1. Survival curves of first sexual experience among all participants and stratified by sex.**

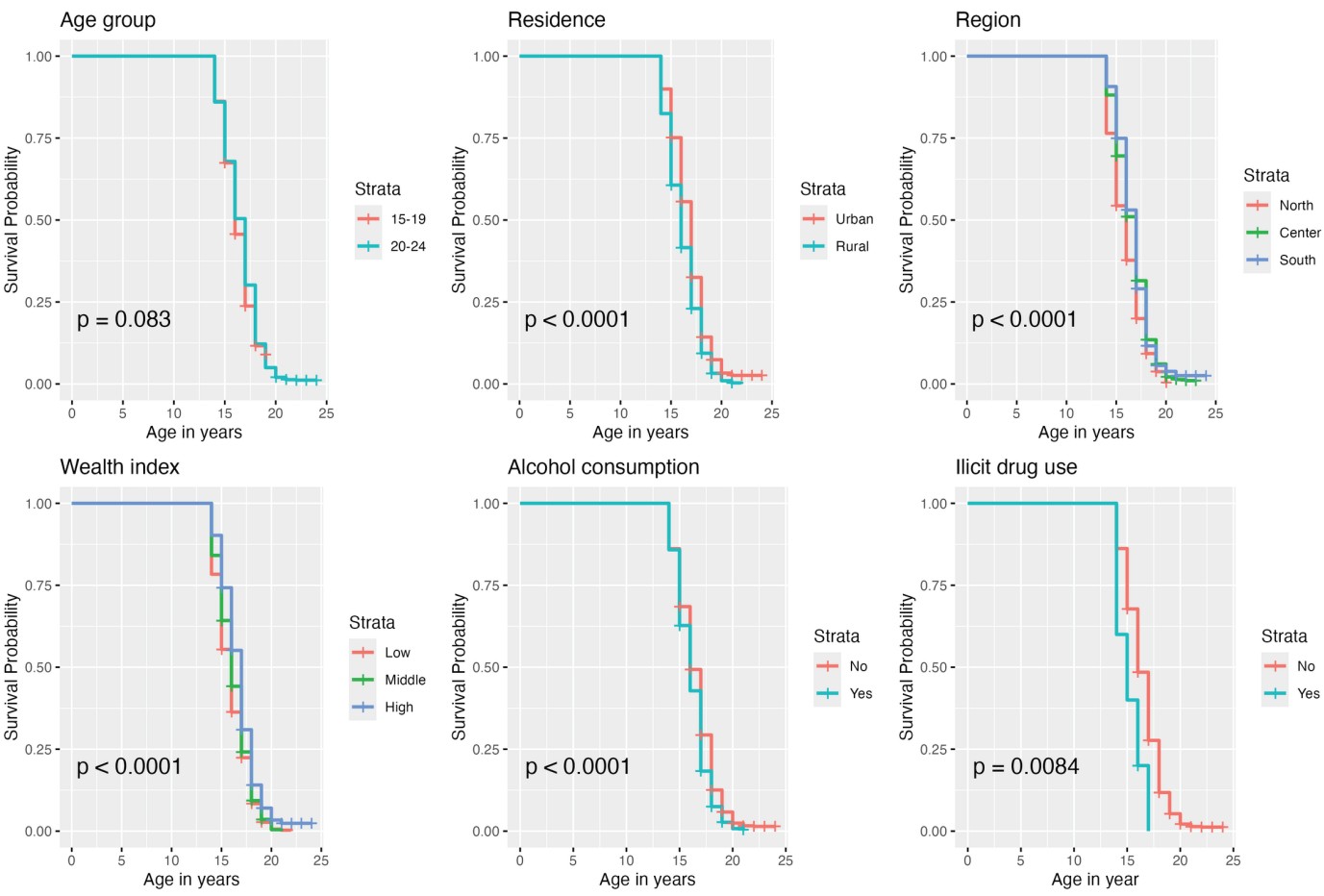

**Fig 2. Survival curves of first sexual experience among female individuals, stratified by selected variables.**

**Kaplan-Meier survival analysis for male individuals.** Fig 3 illustrates the Kaplan-Meier survival curves of sexual debut among male individuals, stratified by selected socio-demographic variables. The survival curves for age group, residence, wealth index, and illicit drug use do not exhibit significant variation across the strata, except for region and alcohol consumption, where significant differences are observed. The findings indicate that males from the southern and central regions are more likely to remain sexually inactive as they age. Additionally, males who reported alcohol consumption have a lower probability of remaining sexually inactive compared to those who do not consume alcohol.

## Covariate effects

Table 2 presents the parameter estimates for both the standard Cox and geoadditive Cox regression models, where we present the overall geoadditive Cox regression model and the stratified models (the geoadditive Cox model is fitted to female and male individuals separately). The estimates from both standard and overall geoadditive Cox model are largely consistent in terms of magnitude and direction. The findings indicate that the hazard of sexual debut for individuals aged 20-24 is 20% lower (aHR = 0.8, 95% CI: 0.71 - 0.82) compared to

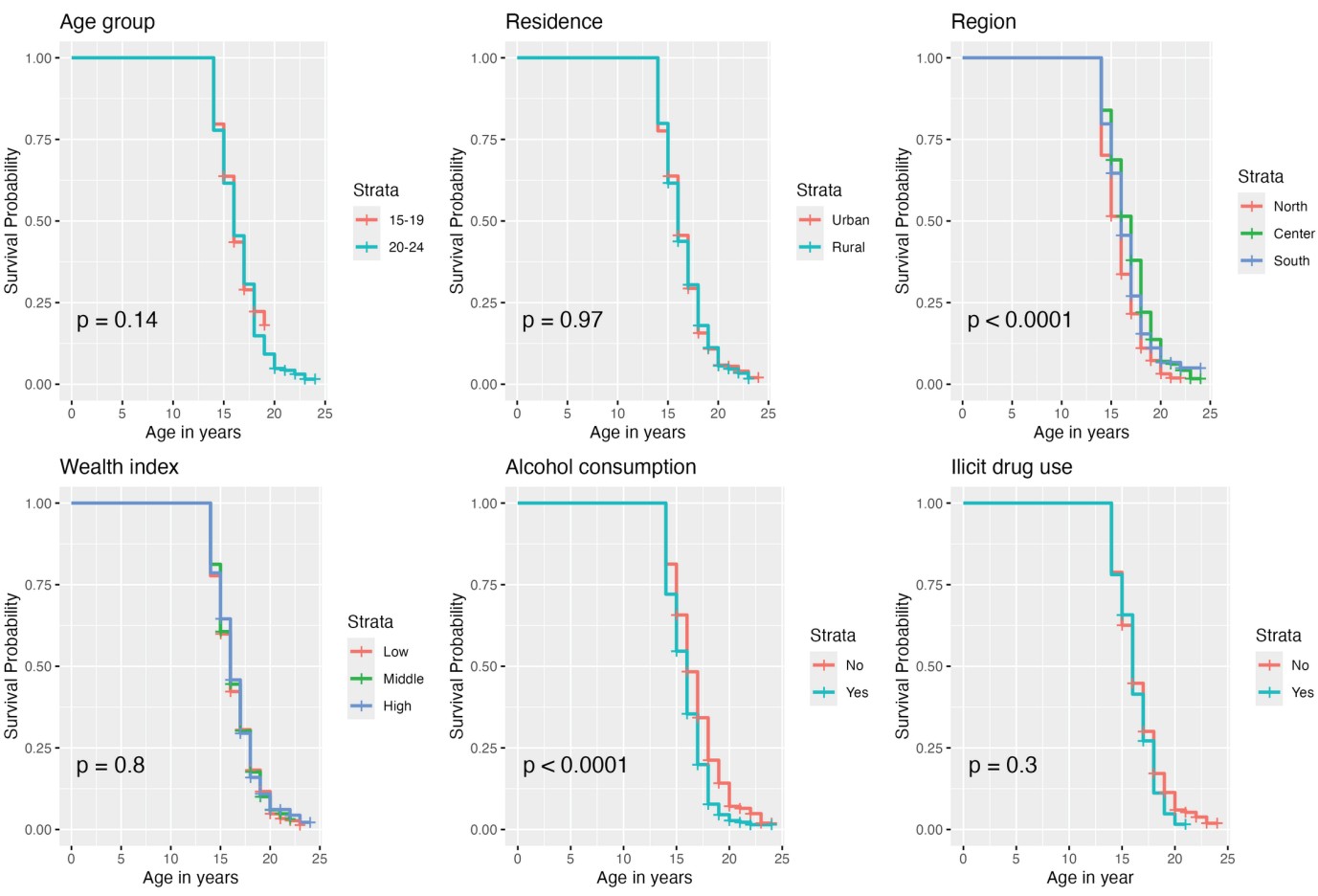

**Fig 3. Survival curves of first sexual experience male individuals, stratified by selected variables.**

those aged 15-19. Similarly, both female and male individuals aged 20-24 exhibit a reduced risk of first sexual experience at earlier age.

The findings presented in Table 2 further indicate that individuals with higher and secondary education levels have a 30% (aHR = 0.7, 95% CI: 0.53 - 0.86) and 20% (aHR = 0.9, 95% CI: 0.74 - 0.96) lower risk of initiating sexual activity at an early age, respectively, compared to those without education. For females, those with some level of education are less likely to experience their first sexual encounter at an early age than those without any education. In contrast, education does not significantly impact the age of first sexual experience among males. Additionally, the results reveal that married individuals are 70% more likely to have an earlier sexual debut compared to single adolescents and youth. This trend is consistent for both married females and males, who exhibit a higher risk of sexual debut at an early age than their single counterparts. Furthermore, individuals with employment have a 10% higher risk of sexual debut at an earlier age compared to those without a job (aHR = 1.1, 95% CI: 1.02 - 1.18). Among females, occupation does not significantly affect the age of sexual debut, whereas among males, employed individuals have a 10% higher risk of experiencing first sex at an early age compared to unemployed males (aHR = 1.1, 95% CI: 1.02 - 1.25).

The risk of sexual initiation at an early age among adolescents and youths who reported alcohol consumption was 50% higher compared to those who did not consume alcohol

**Table 2. Geoadditive Cox proportional hazard regression for adolescents and youth.** The parameters estimates herein presented are the adjusted hazard ratio (aHR) accompanied with the 95% confidence interval (CI). All significant coefficients are emphasized.

| Covariates | Standard Cox model | Geoadditive Cox model | Female geoadditive Cox model | Male geoadditive Cox model |
|---|---|---|---|---|
| | aHR (95% CI) | aHR (95% CI) | aHR (95% CI) | aHR (95% CI) |
| **Age** (ref = 15-19) | | | | |
| 20-24 | **0.7 (0.66 - 0.78)** | **0.8 (0.71 - 0.82)** | **0.7 (0.61 - 0.74)** | **0.8 (0.73 - 0.91)** |
| **Sex** (ref = Male) | | | | |
| Female | 1.0 (0.92 - 1.08) | 1.0 (0.93 - 1.07) | | |
| **Residence** (ref = Urban) | | | | |
| Rural | 1.0 (0.92 - 1.12) | 1.0 (0.94 - 1.11) | 1.1 (1.0 - 1.23) | 1.0 (0.86 - 1.13) |
| **Education** (ref = No education) | | | | |
| Primary | **0.9 (0.75 - 0.99)** | 0.9 (0.80 - 1.01) | **0.9 (0.74 - 0.99)** | 1.0 (0.85 - 1.29) |
| Secondary | **0.8 (0.67 - 0.92)** | **0.8 (0.74 - 0.96)** | **0.7 (0.58 - 0.8)** | 1.2 (0.95 - 1.46) |
| Higher | **0.7 (0.53 - 0.89)** | **0.7 (0.53 - 0.86)** | **0.4 (0.31 - 0.61)** | 1.2 (0.82 - 1.70) |
| **Wealth index** (ref = Low) | | | | |
| Middle | 1.0 (0.85 - 1.06) | 0.9 (0.86 - 1.04) | 0.9 (0.82 - 1.06) | 1.0 (0.83 - 1.13) |
| Higher | 1.0 (0.87 - 1.10) | 1.0 (0.87 - 1.07) | 1.0 (0.90 - 1.17) | 1.0 (0.88 - 1.22) |
| **Marital status** (ref = Single) | | | | |
| Married | **1.8 (1.63 - 1.95)** | **1.7 (1.54 - 1.82)** | **1.8 (1.65 - 2.05)** | **1.4 (1.27 - 1.65)** |
| Divorced/Widowed | **1.7 (1.48 - 1.91)** | **1.6 (1.37 - 1.76)** | **1.6 (1.39 - 1.89)** | **1.4 (1.12 - 1.75)** |
| **Occupation** (ref = Unemployed) | | | | |
| Employed | 1.0 (0.95 - 1.12) | **1.1 (1.02 - 1.18)** | 1.0 (0.94 - 1.15) | **1.1 (1.02 - 1.25)** |
| **Alcohol consumption** (ref = No) | | | | |
| Yes | **1.6 (1.42 - 1.70)** | **1.5 (1.39 - 1.63)** | **1.5 (1.36 - 1.71)** | **1.6 (1.43 - 1.79)** |
| **Illicit drug use** (ref = No) | | | | |
| Yes | 0.8 (0.62 - 1.10) | 0.9 (0.71 - 1.14) | **2.8 (1.47 - 5.45)** | 0.8 (0.66 - 1.09) |
| **Ever talked with parents about sex** (ref = Yes) | | | | |
| No | 0.9 (0.84 - 1.04) | 1.0 (0.87 - 1.06) | 0.9 (0.79 - 1.02) | 1.0 (0.85 - 1.17) |
| **Ever talked with parents about HIV** (ref = Yes) | | | | |
| No | 1.0 (0.91 - 1.12) | 1.0 (0.91 - 1.1) | 1.0 (0.88 - 1.12) | 0.9 (0.72 - 1.21) |
| **Ever attended any HIV prevention program** (ref = Yes) | | | | |
| No | 0.9 ( 0.78 - 1.10) | 0.9 (0.75 - 1.05) | 0.9 (0.7 - 1.08) | 0.8 (0.75 - 0.93) |
| **Ever lived away from home for more than one month** (ref =Yes) | | | | |
| No | **0.9 (0.85 - 0.99)** | **0.9 (0.85 - 0.98)** | 1.0 (0.87 - 1.07) | **0.8 (0.75 - 0.93)** |
| **Latitude** | **1.1 (1.07 - 1.13)** | **1.0 (1.01 - 1.03)** | **1.0 (0.97 - 1.00)** | **1.0 (1.02 - 1.06)** |
| **Longitude** | **1.1 (1.03 - 1.08)** | **1.0 (1.02 - 1.05)** | **1.1 (1.05 - 1.09)** | **1.1 (1.03 - 1.09)** |

(aHR = 1.5, 95% CI: 1.39 - 1.63). This effect was consistent across both female and male individuals. Regarding the use of illicit drugs, the findings indicate that the effect was not statistically significant in the overall model and the male geoaddative model. However, we notice that among female, the risk of first sex at an early age was nearly three times higher for females who reported illicit drug use in the past 12 months (aHR = 2.8, 95% CI: 1.47 - 5.45). Additionally, the results show that individuals who reported not being away from home for more than one month had a 10% lower risk of early sexual initiation compared to those who did (aHR = 0.9, 95% CI: 0.85 - 0.98), with this effect being similar among males.

## Spatial trend

Given the interest in examining the spatial distribution of the hazard of first sex, Fig 4 presents the spatial variation, suggesting a discernible trend in the data for all models. After adjusting for important covariates, the hazard of experiencing first sex at an earlier age is notably higher in the northern and southern regions of the country. Additionally, the

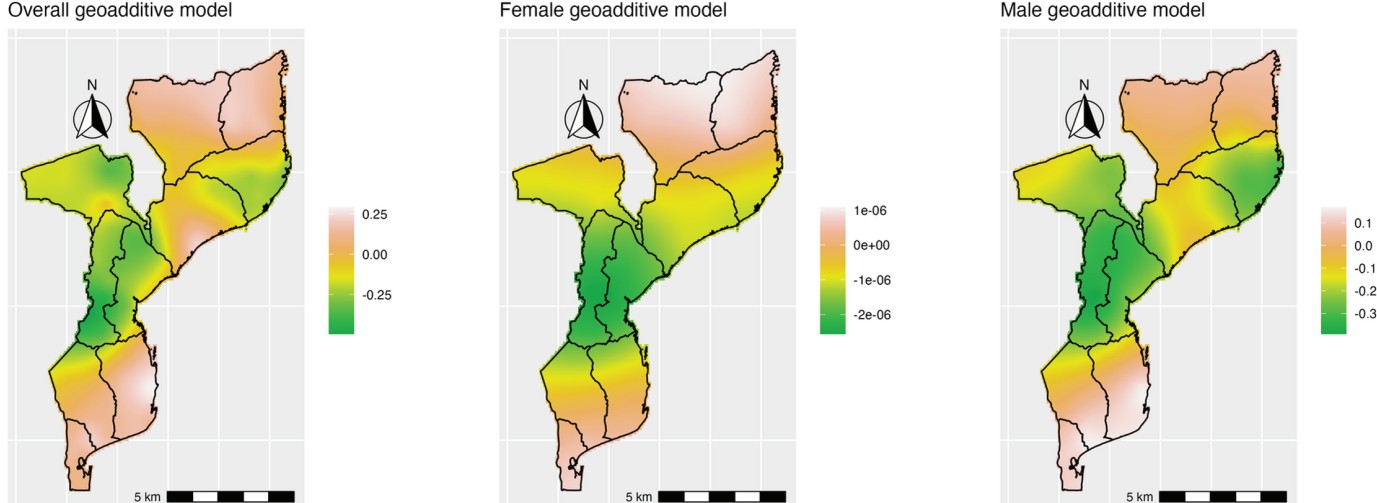

**Fig 4. Radial basis spline plotted on the natural log(HR) scale.** The map shows the spatial variation in the risk of early sexual debut among adolescents and youth in Mozambique

province of Zambézia presents a higher hazard of sexual debut at an earlier age. For females, the results reveal a similar pattern, showing an increased risk of early sexual initiation in some parts of Tete, Zambézia, and Nampula provinces. For males, the map also indicates a higher hazard of first sex at an early age in the southern and northern regions of the country.

## Discussion

The findings of this study underscore the critical public health issue of early sexual debut among adolescents and youth in Mozambique. Our study revealed that the time to sexual debut is associated with a range of socio-demograhic, economic, and behavioral factors. The results revealed that after accounting for important covariates, significant regional disparities emerged, with the hazard of sexual debut being more pronounced in the northern and southern regions, particularly among young females. This can be attributed to the high prevalence of child marriage in these areas. Recent data from United Nations Children's Fund (UNICEF) show that child marriage rates exceed 50% in these regions [35]. Additionally, the increased risk of initiating sexual activity early may be coupled with the ongoing displacements in the northern region due to terrorism, which heightens the vulnerability of girls and boys to sexual assault and forced marriage [36], while in the Gaza and Inhambane provinces, located in the southern region, the elevated risk of initiating sexual activity at an earlier age may be explained by high rates of poverty in this region, as economic hardships can lead to early sexual initiation as adolescents may engage in transactional sex to meet their basic needs [37]. In fact, an assessment conducted by the Ministry of Economy and Finance revealed that the poverty rates is 49.1% in Gaza and 50.8% Inhambane province [38]. This geographic variance underscores the importance of regional-specific health interventions that address the cultural and socioeconomic factors unique to each area.

The results revealed a notable generational shift in the age of sexual initiation between two age groups, 15–19 and 20–24 years, consistent across both genders. This shift suggests changing in social and cultural factors influencing sexual behaviour over time. Furthermore, our analysis identified education as a significant determinant, with adolescents and young

adults lacking formal education being more likely to initiate sexual activity at a younger age than their more educated peers. These findings align with previous research, including studies by [39] in Gambia and [40] in Nigeria. The delay in sexual initiation among individuals with higher levels of education, particularly females, could be attributed to increased awareness of the risks associated with early sexual activity [6]. Education likely contributes to better health outcomes by fostering future-oriented thinking and awareness of the risks associated with early sexual activity. In particular, young women with access to education may delay sexual initiation to avoid disruptions, such as unintended pregnancies or sexually transmitted infections (STIs), allowing them to prioritize academic achievements [39,41].

We also found that marital status plays a crucial role in early sexual debut, with individuals who are married or have previously been married reporting higher rates of early sexual initiation. This finding is consistent with other studies, which show that early marriage often accelerates sexual initiation due to cultural and social expectations surrounding marriage. In regions with prevalent early marriage practices, such as in parts of sub-Saharan Africa, adolescents are likely to enter sexual relationships earlier as marriage formalizes and legitimizes sexual activity at a young age [3,42]. These results highlight the influence of marital norms on sexual debut, particularly in Mozambique's northern and central regions, where child marriage rates are notably high [10,35]. While early marriage is often linked to earlier sexual initiation, it is important to recognize that this relationship may also be shaped by reverse causality. For instance, adolescents who initiate sexual activity early may be more likely to marry young [11]. Additionally, early marriage not only leads to earlier sexual activity but can also increase health risks for young females, including exposure to STIs, unintended pregnancies, and reduced educational attainment [12].

Employment status emerged as another important factor, with employed males being more prone to earlier sexual initiation compared to their unemployed counterparts. This finding echoes research from India by [43], which showed a similar trend among unmarried adolescents. Employment may expose youth to adult environments, leading to diminished parental supervision, increased financial independence, and a sense of maturity that encourages adult behaviors, including sexual activity [44]. Moreover, adolescents in the workforce may be exposed to adult role models who engage in sexual risk behaviors, as suggested by [45]. Furthermore, employment provides adolescents with financial resources that may facilitate participation in unsupervised activities, thereby increasing the likelihood of sexual risk-taking and the potential for adverse health outcomes [46,47]

Finally, our study identified alcohol consumption and illicit drug use as significant predictors of the timing of sexual debut. Adolescents who reported alcohol or illicit drug use were more likely to initiate sexual activity at a younger age. This finding is consistent with [48], who observed that alcohol consumption significantly increases the likelihood of early sexual debut. The relationship between substance use and the time to sexual initiation may be explained by the impairing effects of alcohol and drugs on decision-making and inhibition, leading to risky sexual behaviours. However, it is important to note the potential for reverse causality in this relationship. While substance use may contribute to earlier sexual initiation, it is also plausible that early sexual debut could lead to increased substance use later in life, as a coping mechanism or due to changes in peer networks and social behaviors [49]. The study further identified that the hazard of early sexual debut is reduced among individuals who reported to never have stayed away from their parents, which corroborate with several studies [50,51]. This could be explained by the fact that being close to the parents has a protective role as it provides a protective environment that reduces the likelihood of early sexual initiation [52].

Although this study provides valuable insights into the factors influencing the timing of sexual debut and its spatial heterogeneity among adolescents and youth in Mozambique, several limitations should be acknowledged. First, the geographical coordinates were assigned at the enumeration area level, assuming that all households within a given area share the same location. This simplification may introduce spatial imprecision. Some studies recommend treating this limitation as a form of measurement error by modeling the location as a distribution within each enumeration area [53]. Additionally, the radial basis function used in the spatial model assumes isotropy, whereas many geostatistical processes are anisotropic in nature. Future research could explore the use of anisotropic covariance functions to better capture spatial variation.

Second, the Cox model approximation to Poisson regression relies on the assumption of equidispersion, which was not tested in this study. Future work should assess the robustness of the model under conditions of overdispersion or underdispersion. Similarly, the Poisson approximation assumes a constant hazard ratio, and its validity should be evaluated when this assumption does not hold.

Third, the cross-sectional design of the study limits causal inference. While associations between variables can be identified, the temporal ordering of events cannot be established. The explanatory variables in the present study reflect current conditions at the time of the survey, which may have occurred after the sexual debut. Although these variables can serve as proxies for earlier life circumstances, the possibility of reverse causality should not be overlooked. For instance, early sexual initiation may influence subsequent behaviors such as substance use or early marriage, rather than the reverse. These bidirectional relationships can add complexity in the interpretation and underscore the need for longitudinal studies to clarify causal pathways.

Finally, the reliance on self-reported data introduces the potential for bias, as participants may underreport or overreport sensitive behaviors due to social desirability or recall limitations. As such, the findings should be interpreted with appropriate caution.

## Conclusion

This study highlights the complex interplay of individual, social, and geographic factors contributing to early sexual debut among adolescents and young adults in Mozambique. Our findings reveal significant regional disparities, with a notably higher risk of initiating sexual activity at an earlier age being evident in the northern and southern regions. Education appears to delay sexual debut, while early marriage and employment, especially among young females and working males, increase the likelihood of early sexual activity. Substance use, including alcohol and drugs, also strongly correlates with early initiation, highlighting the need for prevention efforts. These findings have important implications for public health. Interventions should focus on regions where early marriage, low education levels, and substance use are common. Programs that promote education, support youth employment responsibly, and prevent substance use may help reduce early sexual initiation and related health risks. Our results highlight the importance of region-specific, multifaceted approaches to public health that consider education, marital practices, employment, and substance use. Such strategies may offer promising pathways for improving health outcomes among Mozambican youth.

## Acknowledgments

The authors would like to express their sincere gratitude to the Population-based HIV Impact Assessment (PHIA) project for providing the Mozambique Population-based HIV Impact

Assessment (INSIDA) 2021 data used in this analysis. The data provided by PHIA were instrumental in conducting this research and deriving meaningful insights into the factors associated with early sexual debut among adolescents and youth in Mozambique. Special thanks to the reviewers and the editor for their constructive feedback, which greatly improved the quality of the manuscript.

## Author contributions

**Conceptualization:** Rachid Muleia.

**Data curation:** Rachid Muleia, Shelsea Luís Damião, Isaac Akpor Adjei.

**Formal analysis:** Rachid Muleia, Áuria Ribeiro Banze, Cynthia Semá Baltazar, Isaac Akpor Adjei.

**Investigation:** Rachid Muleia.

**Methodology:** Rachid Muleia.

**Software:** Rachid Muleia, Shelsea Luís Damião, Isaac Akpor Adjei.

**Supervision:** Cynthia Semá Baltazar.

**Visualization:** Rachid Muleia.

**Writing – original draft:** Rachid Muleia, Isaac Akpor Adjei.

**Writing – review & editing:** Rachid Muleia, Shelsea Luís Damião, Áuria Ribeiro Banze, Cynthia Semá Baltazar, Isaac Akpor Adjei.

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
