## [Decision Letter · Decision Letter 0]

13 Feb 2025

PONE-D-24-50483Factors associated with early sexual debut among adolescents and youth in Mozambique: a geospatial analysis of the Mozambique  2021 AIDS surveyPLOS ONE

Dear Dr. Muleia,

Thank you for submitting your manuscript to PLOS ONE. After careful consideration, we feel that it has merit but does not fully meet PLOS ONE’s publication criteria as it currently stands. Therefore, we invite you to submit a revised version of the manuscript that addresses the points raised during the review process.

We look forward to receiving your revised manuscript.

Kind regards,

Enamul Kabir

Academic Editor

PLOS ONE

Journal Requirements:

2. In the online submission form, you indicated that the survey data set is publicly available at https://phia.icap.columbia.edu/surveys/ upon reasonable request. For the current study, we considered a sub-sample of 5283 adolescents and youth aged 15-24 who consented to participate. 

3. We note that Figure 4 in your submission contain [map/satellite] images which may be copyrighted. All PLOS content is published under the Creative Commons Attribution License (CC BY 4.0), which means that the manuscript, images, and Supporting Information files will be freely available online, and any third party is permitted to access, download, copy, distribute, and use these materials in any way, even commercially, with proper attribution. For these reasons, we cannot publish previously copyrighted maps or satellite images created using proprietary data, such as Google software (Google Maps, Street View, and Earth). For more information, see our copyright guidelines: http://journals.plos.org/plosone/s/licenses-and-copyright.

a. You may seek permission from the original copyright holder of Figure 4 to publish the content specifically under the CC BY 4.0 license. 

4. Please include a copy of Table 2 and 4 which you refer to in your text on page 9.

Reviewers' comments:

Reviewer's Responses to Questions

**Comments to the Author**

1. Is the manuscript technically sound, and do the data support the conclusions?

Reviewer #1: Yes

Reviewer #2: Yes

2. Has the statistical analysis been performed appropriately and rigorously? 

Reviewer #1: Yes

Reviewer #2: I Don't Know

3. Have the authors made all data underlying the findings in their manuscript fully available?

Reviewer #1: No

Reviewer #2: No

4. Is the manuscript presented in an intelligible fashion and written in standard English?

Reviewer #1: Yes

Reviewer #2: Yes

5. Review Comments to the Author

Reviewer #1: Some comments/questions below that I thought would be worth raising in case minor revisions would be of value for the author to consider.

1. Given that a time to event approach is being used, some of the explanatory variables that are noted appeared to potentially be related to periods that occurred AFTER the event. For example, education, marriage, alcohol & drugs, living away from home etc. While the discussion section does acknowledge that the survey is cross-sectional rather than longitudinal, does the fact that some of these variables are only relevant outside of the hazard period raise a broader methodological question? The section talking about the link between marriage and early sexual debut for example got me thinking about what the relationship was between the sexual debut prior to 15 years and the fact that the marriage may be many years later and potentially unrelated to the sexual partner involved in the earlier event. Similar considerations are at play for employment - is this potentially many years after the early sexual debut rather than prior to the event? Perhaps this could be noted explicitly and/or made clear if these variables are actually all within the pre-event window?

2. There are a couple of small typos or things to check - e.g. the formular for the exponential and gaussian covariance functions are the same, the phase "likewise Ganguli and Wand" may be better as "Similar to...", the label under figure 4 says "you" instead of "youth".

3. In Ganguli & Wand, there was detail provided in relation to the number of knots used within the geospatial smoothing, however the current manuscript does not discuss this detail. The Discussion section however does mention that the 'geographical locations' are the enumeration level - is that what is being used for the knots, rather than a low-rank approach using only a subset of knots? (please excuse if I have misunderstood something here)

4. In Ganguli & Wand, the contour plot of the geospatial factor focused on the log hazard ratio in order to draw attention to where geography seemed to impact the hazard ratio. Would this be worth considering in Figure 4? It's not quite clear what the contour scale is showing at the moment.

5. The socio-demographic profile notes that early debut is significantly more likely for males than females, but the Kaplan-Meier section immediately following calls out that this difference is not significant. Does that warrant a highlight?

6. Similar to #5, there are several references that appear to be conflicting in terms of whether northern/central/southern regions have higher vs lower hazard - e.g. the prevalence rates in table 1 show the northern with a higher prevalence, similar with the Kaplan-Meier results, however the cox model results in table 3 suggest that the northern region (which is the reference) has a lower hazard rate than the others.

Note: I have answered "no" to the data being available, as it seemed that only the raw survey data would be available and the reviewer question seemed to suggest that 'all data underlying the findings' (e.g. 'datapoints behind means') being made available.

Reviewer #2: Thank you for the opportunity to review “Factors associated with early sexual debut among adolescents and youth in Mozambique: a geospatial analysis of the Mozambique 2021 AIDS survey”. The authors data from a national survey to describe an important issue. In general, this paper is clear and well written. However, the authors seem to struggle between presenting results, and defending their modeling strategy in great detail. As a public health epidemiologist, I would encourage the authors to focus on the former, and consider a second methods paper to consider the latter. Below are a few suggestions that I feel may help to improve the paper.

Abstract: Currently, the abstract talks in broad generalities, with no specifics. I appreciate why the authors might have done this, but I ask they consider at least a few numbers. What is the N of their study? What is the overall % of early sexual debut (or, if they wish, the regional % given the assertion of differences)? You don’t need to provide too many details (I think it’s fine, for example, to list correlates, without presenting your measures of effect), but at least tell us the size of your study, and your overall estimate(s).

Abstract: I recommend that you specify the ages of your adolescent and youth data set (i.e., ages 15-24)

Methods, line 79: Why did you consider a subset and not the full data set? How did you select your subset? Your subset represents what percentage of the whole data set?

Methods: can you make your R code and data set available as supplemental material? Your analysis is an interesting one, and I’m sure some readers may be intrigued to review your work and learn more.

Results: Minor point, but “had sex” should probably be “reported having had sex” or similar. Did you do anything to consider reporting bias? Younger participants may have been less confident or comfortable discussing sex and sexual debut, and may have been less likely to report sexual activity.

Results, line 220: Please be cautious with statements like “the prevalence is marginally higher” as the prevalence is statistically equivalent. I do not recommend making claims of modest or marginal difference, where this is really not the case (to me, a “modest” difference is a small, but statistically significant difference, where what you’re showing is very clearly equivalent prevalence across both groups). Kindly go through your paper and consider removing any other claims of this nature.

Results, line 222: How do you define “high” vs “low” socio-economic status? I could not find this specified in your methods.

Results line 242-3: I thought your outcome was early sexual debut, not just sexual debut? This sentence wasn’t very clear to me. Your methods make clear that your outcome is not sexual debut, yes/no, but rather age at sexual debut.

Results, model selection. This section is very technical, and I’m not sure it brings much to the discussion of sexual debut. What is the importance/rational for comparing “circular, exponential, Gaussian, multiquadratic inverse, Mat´ern, and spherical covariance functions” and what is this telling us about age of sexual debut in Mozambique? Some of this section may be best suited to the discussion, or to a more detailed methods paper to describe why you selected the model you did. I don’t think this belongs here in the results, however – or perhaps in a less technical, distilled sentence or two. (Same with Table 2)

Results, “Covariate effects”: Are you modeling “sexual debut” as a bivariate, yes/no, outcome (e.g., lines 281, 283, 284, etc)? That’s how this section reads. However, your methods section makes it sound like you are modeling time to sexual debut? Or early sexual debut (i.e., sex before age 15), yes/no? Please clarify.

Table 3: similar to above, I recommend clearly stating your outcome in the table header. Time to sexual debut? Report of sex before the age of 15 (i.e., yes/no)?

Results, spatial trend: I would encourage the authors to avoid technical jargon in the results. It’s good to explain what you did in the methods, but in the results section, I recommend stating what you found (only). I can refer back to the methods if I wish for more details. (i.e., edit to remove “visualization of the radial basis spline” and “overall geoadditive model” etc)

Discussion: The authors seem torn between presenting their results, and discussing the merits of their modeling approaches. The whole first paragraph of your results re-states your modeling strategy! If I’m hear to learn about early sexual debut in Mozambique, I likely have little interest (or background to understand) why you adopted one strategy over another. I think you may have two papers, here, a methods paper (where you can go much deeper) and a results paper (where you really should limit much of your discussion of methods). Start your discussion with a high level summary of what you found (ie., remove the first paragraph of the discussion –I am guessing most with an interest in the topic would want to start with your second paragraph).

6. PLOS authors have the option to publish the peer review history of their article (what does this mean?). If published, this will include your full peer review and any attached files.

Reviewer #1: No

Reviewer #2: **Yes: **Matt Price

---

## [Author Response · Author response to Decision Letter 1]

31 Mar 2025

Reviewer 1

1. Given that a time to event approach is being used, some of the explanatory variables

that are noted appeared to potentially be related to periods that occurred AFTER

the event. For example, education, marriage, alcohol & drugs, living away from home

etc. While the discussion section does acknowledge that the survey is cross-sectional

rather than longitudinal, does the fact that some of these variables are only relevant

outside of the hazard period raise a broader methodological question? The section

talking about the link between marriage and early sexual debut for example got me

thinking about what the relationship was between the sexual debut prior to 15 years

and the fact that the marriage may be many years later and potentially unrelated to

the sexual partner involved in the earlier event. Similar considerations are at play for

employment - is this potentially many years after the early sexual debut rather than

prior to the event? Perhaps this could be noted explicitly and/or made clear if these

variables are actually all within the pre-event window.

1

It’s true that these variable may be potentially related to the periods that occurred

after the event. Additionally, from the survey data, it is not possible to tell weather

they are within the pre-event window. Nevertheless, they serve as proxies for earlier

life circumstances and behaviors, which can actually help us have some sense about

the past conditions based on the current status.

For example, the current education level may reflect the individual’s educational trajectory

and access to education during their formative years, which, for instance, if

the individual’s current education is secondary, that may suggest that during the their

adolescence they were engaged in school, and this may have influenced to delay their

sexual debut. Similarly, the marital status can indicate social and cultural contexts

that might have influenced early sexual debut. In many societies, the early marriage if

often linked to early sexual debut. So, this can give us some idea on social environment

of the participant during adolescence.

Regarding employment status, it can reflect socio-economic conditions and responsibilities

that might have existed during adolescence. For example, if an individual is

currently employed, it may suggest they had to assume adult roles and responsibilities

early, potentially correlating with early sexual debut.

We recognize that some of these variables may not fall within the pre-event window,

which is a limitation of the study, and the results should be interpreted with caution.

However, this does not diminish their relevance in explaining early sexual debut,

keeping in mind that they are proxies. This has been stated in the manuscript.

2. There are a couple of small typos or things to check - e.g. the formular for the exponential

and gaussian covariance functions are the same, the phase ”likewise Ganguli and

Wand” may be better as ”Similar to...”, the label under figure 4 says ”you” instead of

”youth”.

2

Thank you for this comment. We have corrected the errors.

3. In Ganguli & Wand, there was detail provided in relation to the number of knots used

within the geospatial smoothing, however the current manuscript does not discuss this

detail. The Discussion section however does mention that the ’geographical locations’

are the enumeration level - is that what is being used for the knots, rather than a lowrank

approach using only a subset of knots? (please excuse if I have misunderstood

something here)

Thank you for this observation. We used a low rank approach and with have clearly

stated this in the manuscript.

4. In Ganguli & Wand, the contour plot of the geospatial factor focused on the log hazard

ratio in order to draw attention to where geography seemed to impact the hazard ratio.

Would this be worth considering in Figure 4? It’s not quite clear what the contour

scale is showing at the moment.

Thank you for this observation. We have also focused on the log hazard ratio scale.

We have mention this in the manuscript.

5. The socio-demographic profile notes that early debut is significantly more likely for

males than females, but the Kaplan-Meier section immediately following calls out that

this difference is not significant. Does that warrant a highlight?

Indeed, we noted that early sexual debut is more common among males. It is important

to emphasize that the variable “early sexual debut” is binary, and the test used is the

Pearson Chi-square, which is sensitive to differences in proportions. These differences

can occur due to study imbalance. In contrast, the Kaplan-Meier analysis examines

the time to sexual debut, and the log-rank test used is sensitive to differences in the

timing of the event. The non-significant result in the Kaplan-Meier analysis indicates

that when considering the entire range of ages for sexual debut, the timing does not

differ significantly between genders. Additionally, the sociodemographic profile was

presented as exploratory to provide more context about the data.

3

6. Similar to 5, there are several references that appear to be conflicting in terms of

whether northern/central/southern regions have higher vs lower hazard e.g. the prevalence

rates in table 1 show the northern with a higher prevalence, similar with the

Kaplan-Meier results, however the cox model results in table 3 suggest that the northern

region (which is the reference) has a lower hazard rate than the others.

Thank you for this observation, it helped us review our analysis. Actually, we noticed

that it was redundant to include the variable region in the model as geographic differences

can be accounted for by the spatial component in the model, which is modeled

through radial basis splines. We have corrected this and all the results are aligned.

Reviewer 2

1. Abstract: Currently, the abstract talks in broad generalities, with no specifics. I

appreciate why the authors might have done this, but I ask they consider at least a

few numbers. What is the N of their study? What is the overall % of early sexual debut

(or, if they wish, the regional % given the assertion of differences)? You don’t need

to provide too many details (I think it’s fine, for example, to list correlates, without

presenting your measures of effect), but at least tell us the size of your study, and

your overall estimate(s). Abstract: I recommend that you specify the ages of your

adolescent and youth data set (i.e., ages 15-24)

Thank you for this suggestion. We have included this information in the Abstract

2. Methods, line 79: Why did you consider a subset and not the full data set? How did

you select your subset? Your subset represents what percentage of the whole data set?

Actually, we wanted to clarify that from the entire data set of individuals aged 15 and

above, we only considered adolescents and youth aged 15 to 24 years, totaling 5,283

individuals. We apologize for any confusion, as the text may have implied that we

considered a sub-sample from individuals aged 15 to 24 years.

4

3. Methods: can you make your R code and data set available as supplemental material?

Your analysis is an interesting one, and I’m sure some readers may be intrigued to

review your work and learn more.

We have made the R code publicly available in a github repository at https://github.

com/RachidMuleia/sexual_debut.git

4. Results: Minor point, but “had sex” should probably be “reported having had sex” or

similar. Did you do anything to consider reporting bias? Younger participants may

have been less confident or comfortable discussing sex and sexual debut, and may have

been less likely to report sexual activity.

Thank you. We have corrected it . With regard to reporting bais, considering this

is a cross-sectional survey, it is expected that some participants might omit some

information. Therefore, the results should interpreted bearing in mind that. We discuss

it as a limitation of the study.

5. Results, line 220: Please be cautious with statements like “the prevalence is marginally

higher” as the prevalence is statistically equivalent. I do not recommend making claims

of modest or marginal difference, where this is really not the case (to me, a “modest”

difference is a small, but statistically significant difference, where what you’re showing

is very clearly equivalent prevalence across both groups). Kindly go through your paper

and consider removing any other claims of this nature.

Thank you for this. We have gone through the text and corrected all such statement

in the text.

6. Results, line 222: How do you define “high” vs “low” socio-economic status? I could

not find this specified in your methods.

Sorry for this. we have properly defined this in the method section.

7. Results line 242-3: I thought your outcome was early sexual debut, not just sexual

debut? This sentence wasn’t very clear to me. Your methods make clear that your

5

outcome is not sexual debut, yes/no, but rather age at sexual debut.

Indeed, the outcome is early sexual debut. We have corrected it every where in the

text. Thank you for this observation.

8. Results, model selection. This section is very technical, and I’m not sure it brings much

to the discussion of sexual debut. What is the importance/rational for comparing “circular,

exponential, Gaussian, multiquadratic inverse, Mat´ern, and spherical covariance

functions” and what is this telling us about age of sexual debut in Mozambique? Some

of this section may be best suited to the discussion, or to a more detailed methods

paper to describe why you selected the model you did. I don’t think this belongs here

in the results, however – or perhaps in a less technical, distilled sentence or two. (Same

with Table 2)

Thank you for this observation. We have removed this section and described the

motivation for choosing spherical semivariogram in the method section.

9. Results, “Covariate effects”: Are you modeling “sexual debut” as a bivariate, yes/no,

outcome (e.g., lines 281, 283, 284, etc)? That’s how this section reads. However, your

methods section makes it sound like you are modeling time to sexual debut? Or early

sexual debut (i.e., sex before age 15), yes/no? Please clarify.

We are modellig time to sexual debut and we have made it clear in the text.

10. Results, spatial trend: I would encourage the authors to avoid technical jargon in the

results. It’s good to explain what you did in the methods, but in the results section, I

recommend stating what you found (only). I can refer back to the methods if I wish for

more details. (i.e., edit to remove “visualization of the radial basis spline” and “overall

geoadditive model” etc)

Thank you for this observation. We have removed the technical jargon in the result

section.

11. Discussion: The authors seem torn between presenting their results, and discussing the

6

merits of their modeling approaches. The whole first paragraph of your results re-states

your modeling strategy! If I’m hear to learn about early sexual debut in Mozambique, I

likely have little interest (or background to understand) why you adopted one strategy

over another. I think you may have two papers, here, a methods paper (where you

can go much deeper) and a results paper (where you really should limit much of your

discussion of methods). Start your discussion with a high level summary of what you

found (ie., remove the first paragraph of the discussion –I am guessing most with an

interest in the topic would want to start with your second paragraph).

Thank you for this observation. We have revomed the first paragraph of the discussion,

focusing more on reporting the summary findings.

Editor

1. Please ensure that your manuscript meets PLOS ONE’s style requirements, including

those for file naming.

We have tried follow all of the journal’s requirements.

2. 2. In the online submission form, you indicated that the survey data set is publicly

available at https://phia.icap.columbia.edu/surveys/ upon reasonable request. For the

current study, we considered a sub-sample of 5283 adolescents and youth aged 15-24

who consented to participate.

We have provided the data and the code used in the manuscript and the data can be

found at (https://github.com/RachidMuleia/sexual_debut.git)

3. We note that Figure 4 in your submission contain [map/satellite] images which may be

copyrighted. All PLOS content is published under the Creative Commons Attribution

License (CC BY 4.0), which means that the manuscript, images, and Supporting Information

files will be freely available online, and any third party is permitted to access,

download, copy, distribute, and use these materials in any way, even commercially, with

7

proper attribution. For these reasons, we cannot publish previously copyrighted maps

or satellite images created using proprietary data, such as Google software (Google

Maps, Street View, and Earth). For more information, see our copyright guidelines:

http://journals.plos.org/plosone/s/licenses-and-copyright.

We require you to either (1) present written permission from the copyright holder to

publish these figures specifically under the CC BY 4.0 license, or (2) remove the figures

from your submission:

With regard to Figure 4, the shapefile used to produce this figure is available at 

https://data.humdata.org/dataset/cod-ab-moz. And according to their police all the

data available at their website is licensed under a Creative Commons Attribution 4.0,

which means that can be freely downloaded and distributed.

4. Please include a copy of Table 2 and 4 which you refer to in your text on page 9.

We have provided table 2. With regard to Table 4 this was a mistake we don’t have

Table 4 in the manuscript. We have corrected this

8

---

## [Decision Letter · Decision Letter 1]

28 May 2025

PONE-D-24-50483R1Factors associated with early sexual debut among adolescents and youth in Mozambique: a geo-additive survival analysis of the Mozambique 2021 AIDS indicator surveyPLOS ONE

Dear Dr. Muleia,

Thank you for submitting your manuscript to PLOS ONE. After careful consideration, we feel that it has merit but does not fully meet PLOS ONE’s publication criteria as it currently stands. Therefore, we invite you to submit a revised version of the manuscript that addresses the points raised during the review process.

The previous editor was not available, so that I have taken their role in the revision. Among the two previous reviewers only reviewer 1 was available, and proposes to accept the manuscript. Reviewer 2's suggestions have been, according to the new editor, taken up.My only concern is with the reaction to the first issue raised by reviewer 1.403-404Moreover, most of the variables analyzed, such as education, marital status, and employment, may pertain to periods following the event of interest (sexual debut). 

I think this shoud be clearer and more comprehensive.

Clearer: The variables measure current conditions at the time of the survey so that they reflect events happening after the sexual debut.

More comprehensive: You have mentioned some of the least problematic variables on that respect. I think there are three variables that are more problematic:

- Sex of household head: I would drop it from the analysis. I do not know what is the rationalle, but note how this would be different in married/divorced from unmarried adolescents living with their parents. It will confound different effects. You are not providing any rationale. I would remove it overall from the study.

- Recent consumption of alcohol / illicit drugs. In this case there could be reversed causality.

- Marital status: You have mentioned this as a factor for early sexual debut and, in this case, you expect a causal effect from early marriage to early sexual debut. It is true that marriage behaviour could also adapt after early sexual debut leading to reverse causality.As commented: make clearer, and more comprehensive. Include the words reverse causality where appropriate.

A marked-up copy of your manuscript that highlights changes made to the original version. You should upload this as a separate file labeled 'Revised Manuscript with Track Changes'.An unmarked version of your revised paper without tracked changes. You should upload this as a separate file labeled 'Manuscript'.

We look forward to receiving your revised manuscript.

Kind regards,

José Antonio Ortega, Ph.D.

Academic Editor

PLOS ONE

Journal Requirements:

Reviewers' comments:

Reviewer's Responses to Questions

**Comments to the Author**

1. If the authors have adequately addressed your comments raised in a previous round of review and you feel that this manuscript is now acceptable for publication, you may indicate that here to bypass the “Comments to the Author” section, enter your conflict of interest statement in the “Confidential to Editor” section, and submit your "Accept" recommendation.

Reviewer #1: All comments have been addressed

2. Is the manuscript technically sound, and do the data support the conclusions?

Reviewer #1: Yes

3. Has the statistical analysis been performed appropriately and rigorously? 

Reviewer #1: Yes

4. Have the authors made all data underlying the findings in their manuscript fully available?

Reviewer #1: Yes

5. Is the manuscript presented in an intelligible fashion and written in standard English?

Reviewer #1: Yes

6. Review Comments to the Author

Reviewer #1: I observed a small number of typographical errors that would be worth fixing if possible - e.g. the figure of 18.9% early debut sex is also quoted as 18.8% elsewhere in the paper. There is also a typo "individual male individual" and the word "analyzes". Small observations.

7. PLOS authors have the option to publish the peer review history of their article (what does this mean?). If published, this will include your full peer review and any attached files.

Reviewer #1: No

---

## [Author Response · Author response to Decision Letter 2]

4 Jun 2025

Revewer 1

I observed a small number of typographical errors that would be worth fixing if pos-

sible - e.g. the figure of 18.9% early debut sex is also quoted as 18.8% elsewhere in

the paper. There is also a typo ”individual male individual” and the word ”analyzes”.

Small observations.

Thank you for this observation, we have corrected it.

Editor

1. Clearer: The variables measure current conditions at the time of the survey so that

they reflect events happening after the sexual debut.

R: Thank you for this observation, we have clearly stated this in the manuscript

2. 2. More comprehensive: You have mentioned some of the least problematic variables

on that respect. I think there are three variables that are more problematic: - Sex of

household head: I would drop it from the analysis. I do not know what is the rationalle,

but note how this would be different in married/divorced from unmarried adolescents

living with their parents. It will confound different effects. You are not providing any

rationale. I would remove it overall from the study.

R: Thank you for this observation, we have removed the variable Sex of the household

from the analysis

Recent consumption of alcohol/illicit drugs. In this case there could be reversed causal-

ity.

R: Thank you for this observation, this was not clear before as we only thought that the

relationship could be in one direction, but now we have a deeper understanding of

reverse causality concept and we have properly mentioned this in the manuscript as

it’s implication in the analysis.

4. Marital status: You have mentioned this as a factor for early sexual debut and, in this

case, you expect a causal effect from early marriage to early sexual debut. It is true

that marriage behaviour could also adapt after early sexual debut leading to reverse

causality.

R: Tank you for this observation, we have addressed this in the manuscript

5. As commented: make clear, and more comprehensive. Include the words reverse causal-

ity where appropriate.

R:Thank you for this observation. We have properly included reverse causality where

needed.

---

## [Editor Report · Decision Letter 2]

6 Jun 2025

Factors associated with early sexual debut among adolescents and youth in Mozambique: a geo-additive survival analysis of the Mozambique 2021 AIDS indicator survey

PONE-D-24-50483R2

Dear Dr. Muleia,

We’re pleased to inform you that your manuscript has been judged scientifically suitable for publication and will be formally accepted for publication once it meets all outstanding technical requirements.

Kind regards,

José Antonio Ortega, Ph.D.

Academic Editor

PLOS ONE

Additional Editor Comments (optional):

In the opinion of this editor, the authors have satisfactorily introduced the changes suggested by one reviewer and the editor on the previous draft.
---

## [Editor Report · Acceptance letter]

PONE-D-24-50483R2

PLOS ONE

Dear Dr. Muleia,

I'm pleased to inform you that your manuscript has been deemed suitable for publication in PLOS ONE. Congratulations! Your manuscript is now being handed over to our production team.

Kind regards,

on behalf of

Dr. José Antonio Ortega

Academic Editor

PLOS ONE